# SHARED GRADIENT DISCOVERY & SUPERPOSITION: LEARNING DYNAMICS OF GENERALIZATION IN LLMS

**Andrei Mircea**[1,2]  **Ildus Sadrtdinov**[3]  **Irina Rish**[1,2]  **Ekaterina Lobacheva**[1,2]
[1]Mila – Quebec AI Institute    [2]University of Montreal    [3]Constructor University
 Correspondence: `mirceara@mila.quebec`

## ABSTRACT

We propose shared gradient discovery & superposition as a mechanism underlying generalization in LLMs, where shared gradients lead to inherently generalizing shared solutions. To validate our hypothesis, we study circuit emergence as one form of learning such generalizing solutions. We find that our hypothesis can indeed explain and shed new light on circuit emergence and generalization.

## 1   INTRODUCTION

**Towards a mechanistic understanding of how LLMs learn to generalize.**
Generalization in large language models (LLMs) is notoriously jagged, exhibiting remarkable capabilities in some contexts while failing unexpectedly in others. Furthermore, improving generalization in LLMs remains an open problem, with most recent progress arguably having been driven by the scaling up of datasets and models. One reason for this limited progress beyond scaling is our limited mechanistic understanding of how LLMs learn to generalize. By understanding a mechanism underlying a phenomenon, we potentially become able to directly target and control it. In other words, a mechanistic understanding how LLMs learn to generalize, which this paper aims towards, becomes a foundation for principled and interpretable methods to improve generalization in LLMs.

**Challenges limiting our mechanistic understanding of how LLMs learn to generalize.**
Typically, mechanistic understanding in the physical sciences is limited by what we can observe and the resolution at which we can measure it. In contrast, the internals of an LLM are in principle fully observable and measurable, but are in practice too high-dimensional to be interpretable. Progress towards a mechanistic understanding of how LLMs learn to generalize is therefore conditioned on finding the right abstractions that can reduce the dimensionality of this problem into mechanisms that are interpretable but also measurable and actionable. One such abstraction for understanding LLM training dynamics is per-example gradient interactions, which Mircea et al. (2025) formalize in terms of destructive interference to show how scaling improves LLMs. Building on this work and the key idea of per-example gradient interactions, we propose and characterize *shared gradient discovery & superposition* as a related mechanism for how LLMs learn to generalize.

**Shared gradient discovery & superposition as a mechanism for generalization in LLMs.**
LLM training examples can have shared underlying solutions that, if learned, allow LLMs to generalize to unseen examples. Intuitively, for LLMs to learn such shared solutions, corresponding examples should have shared (or aligned) gradients during model training. In Section 2, we formalize this intuition and extend it with a notion of superposition (Elhage et al., 2022) that we adapt to gradients to account for compositionality in shared solutions. We then propose a high-level mechanism by which shared gradient discovery could underlie generalization in LLMs . As a first step towards validating this hypothesis, we measure shared gradient discovery in the context of circuit emergence (Tigges et al., 2024). Importantly, circuits can be seen as shared solutions that generalize to unseen examples by implementing specific algorithms, making circuit emergence an ideal testbed for our hypothesis. In Section 3, we show how shared gradient discovery can not only explain but also shed new light on circuit emergence. In particular, we show how circuit emergence progresses in stages of learning and building on progressively more sophisticated shared solutions, each with corresponding shared gradient directions, consistent with our hypothesis and prior work on circuit emergence. More generally, our findings tentatively support our hypothesis that shared gradient discovery underlies LLM generalization, paving the way for future methodologies that directly promote shared gradient discovery & superposition during training to improve generalization in LLMs.

## 2 SHARED GRADIENT DISCOVERY AND GENERALIZATION

In this section, we first define the notion of **shared gradients** used throughout the paper; showing how it inherently relates to generalization, notably in terms of learning shared solutions that generalize to unseen examples. We then introduce the notions of **gradient superposition** and **superposed gradient features** to extend our definition of shared gradients and account for compositionality in shared solutions and generalization. Lastly, we highlight key factors that make shared gradient discovery & superposition particularly likely to underlie generalization in LLMs specifically.

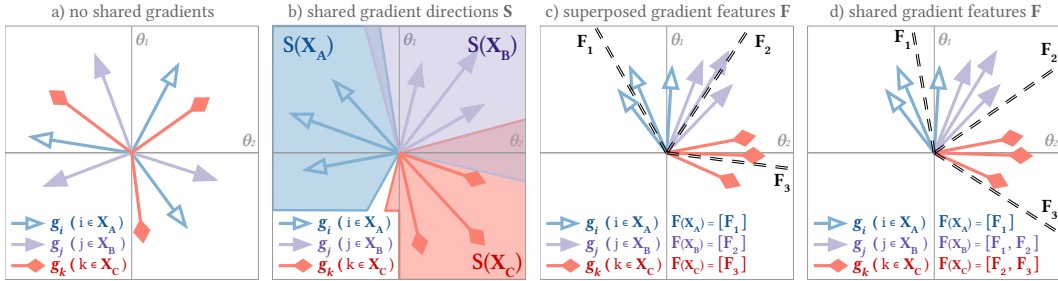

Figure 1: Illustration of shared gradients and superposition in two dimensions.

### SHARED GRADIENTS ARE SHARED SOLUTIONS THAT GENERALIZE.

> **Shared gradients.** Let $i \in X$ be a set of examples with corresponding losses $\ell_i$ and gradients $g_i = \delta\ell_i/\delta\theta$. We say $X$ has **shared gradients** if each example gradient $g_i$ is aligned with every other example gradient $g_{i'\neq i}$, i.e. if dot products $\langle g_i, g_{i'\neq i}\rangle > 0$ for all $i, i' \in X$. When $X$ has shared gradients, we say $S^X$ is a set of **shared gradient directions** for $X$ when $\langle S_j, g_i\rangle > 0$ for all $i \in X$ and $S_j \in S^X$ (Fig. 1b).

Assume losses $\ell_i$ are locally linearizable around $\theta$ such that $\Delta\ell_i = \langle \Delta\theta, g_i\rangle$ for small $\Delta\theta$. Then, gradient descent on any one example $i$, where $\Delta\theta = -\eta g_i$ with small learning rate $\eta$, will also improve on other examples' losses $\ell_{i'\neq i}$ if and only if $\langle g_i, g_{i'}\rangle > 0$. In other words, learning any one example also results in learning every other example with shared gradients, even if not seen in training. More generally, weight updates along shared gradient directions can be seen as learning shared solutions for which loss improvements will inherently generalize to unseen examples.

### SUPERPOSED GRADIENT FEATURES ARE COMPOSITIONAL SHARED SOLUTIONS.

> **Gradient superposition.** In gradient superposition, example gradients $g_i$ can be factorized as linear combinations of near-orthogonal **gradient features** $F_j$ that are sparsely activated across examples (Fig. 1c). Concretely, $g_i = a_{1(g_i)}F_1 + a_{2(g_i)}F_2 + ...$; where $a_{j(g_i)}$ represents how strongly feature $F_j$ is activated. We say $X$ has **shared gradient features** $F^X$ when $\langle F_j, g_i\rangle > 0$ for all $i \in X$ and $F_j \in F^X$ (Fig. 1d).

In the context of gradient superposition, shared solutions can be seen as combinations of gradient features; with solutions that include multiple features, and features that are included in multiple solutions. This factorization is significant as it enables compositionality in shared solutions and generalization. We can extend our definition of shared gradients to account this by viewing $F^X$ as a linear transformation, specifically a matrix that projects gradients onto features $F_j \in F^X$. Concretely, we can say $X$ has shared gradients in $F^X$ when $\langle F^X g_i, F^X g_{i'\neq i}\rangle > 0$ for all $i, i' \in X$. For example, if both $X_A$ and $X_B$ have shared gradients in $F^{X_A}$, then learning any one example in $X_A$ will result in learning shared gradient features $F^{X_A}$ that also generalize to examples in $X_B$.

### SHARED GRADIENT DISCOVERY AS GENERALIZATION IN LLMs.

LLM pretraining is characterized by two considerations which we hypothesize make shared gradients particularly important for generalization in LLMs specifically. First, LLM pretraining is typically single epoch, in the sense that any training example is in principle sampled once throughout training. Second, per-token gradients become systematically opposed during training, making it impossible for models to learn one thing without also unlearning another (Mircea et al., 2025). Taken together, these suggest that generalization in LLMs requires not only discovering shared gradients, but also preserving them throughout training to mitigate gradient destructive interference and prevent unlearning.

## 3 CIRCUIT EMERGENCE IN LLMS AS SHARED GRADIENT DISCOVERY

Circuits are subgraphs of neural networks, where weights connect features to create meaningful algorithms (Olah et al., 2020). Notably, learning such algorithms corresponds to learning shared solutions that generalize to unseen examples, making circuits an ideal testbed for our hypothesis which predicts that shared gradient discovery should underlie circuit learning. In this section we show that shared gradient discovery & superposition can indeed explain, and likely underlies phenomena of circuit emergence identified by Tigges et al. (2024), specifically IOI circuits (Wang et al., 2023). We provide further details and show similar results on additional circuits in Appendix B and C.

**Indirect Object Identification (IOI)** corresponds to e.g. predicting *'Mary'* as the next token in the sentence *'When Mary and John went to the store, John gave a drink to...'.* This task involves logical reasoning to deduce from context that *'Mary'* is the most likely indirect object, and Wang et al. (2020) show that LLMs learn a circuit implementing an algorithmic solution shared across diverse examples. In this section, we use a variant of their dataset with 1000 examples. We refer to this dataset as **[IOI]**, including controls for predicting random tokens **[IOI-rand]**, random names **[IOI-name]**, or previous names in a prompt **[IOI-copy]** to disentangle potential confounds in our analyses.

**Measuring shared gradient discovery** is an open problem with multiple complementary approaches and trade-offs, further discussed in Appendix D.2. In this section, we report mean and standard deviation for pairwise **[IOI]** gradient cosine similarities $\cos(g_i, g_{i' \neq i})$. In our experiments, we use model checkpoints published by Mircea et al. (2025), further described in Appendix D.1.

**RESULT 1:** *Shared gradient discovery occurs with circuit emergence*
Consistent with Tigges et al. (2024), we see in Figure 2 that IOI capabilities emerge concurrently across scales, with loss on IOI predictions improving from step 200-500 onward. At the same time, we see that models indeed discover shared gradients for **[IOI]**. This shared gradient discovery begins before, and continues with, the emergence of IOI; before stabilizing near step 1000 onward. Replicating this analysis on our **[IOI-rand]** control in Appendix B.2, we confirm that these results are specific to IOI and cannot be attributed to shared structure in the prompts independent of IOI.

**RESULT 2:** *Shared gradients are likely necessary for, and underlie, circuit emergence*
Consistent with our hypothesis, shared gradients in **[IOI]** are not only discovered, but also preserved throughout training. In Section 2 we describe how LLM learning dynamics, particularly destructive interference in gradients, can make shared gradients necessary for learning generalizing solutions. In Figure 2, we confirm the key assumption that destructive interference from pretraining gradients occurs for **[IOI]** specifically. As a result of this noisy but systematic destructive interference, it becomes highly unlikely to learn IOI without shared gradients, as random training batches do not lead to consistent improvements in the IOI loss. In other words, discovering and preserving shared gradients not only co-occurs with the emergence of IOI capabilities, but very likely underlies it in terms of training dynamics. Further validating this relationship with theory and experiments remains an important direction for future research.

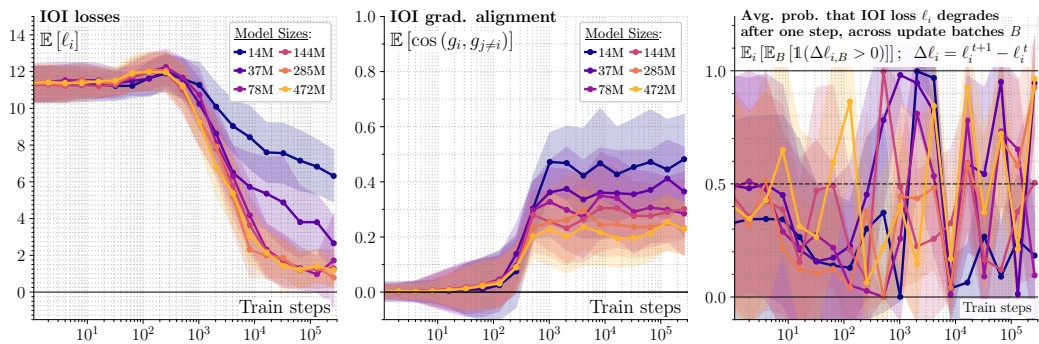

Figure 2: IOI capabilities emerge following shared gradient discovery (left, middle). Destructive interference from training updates make it unlikely to learn IOI without shared gradients (right).

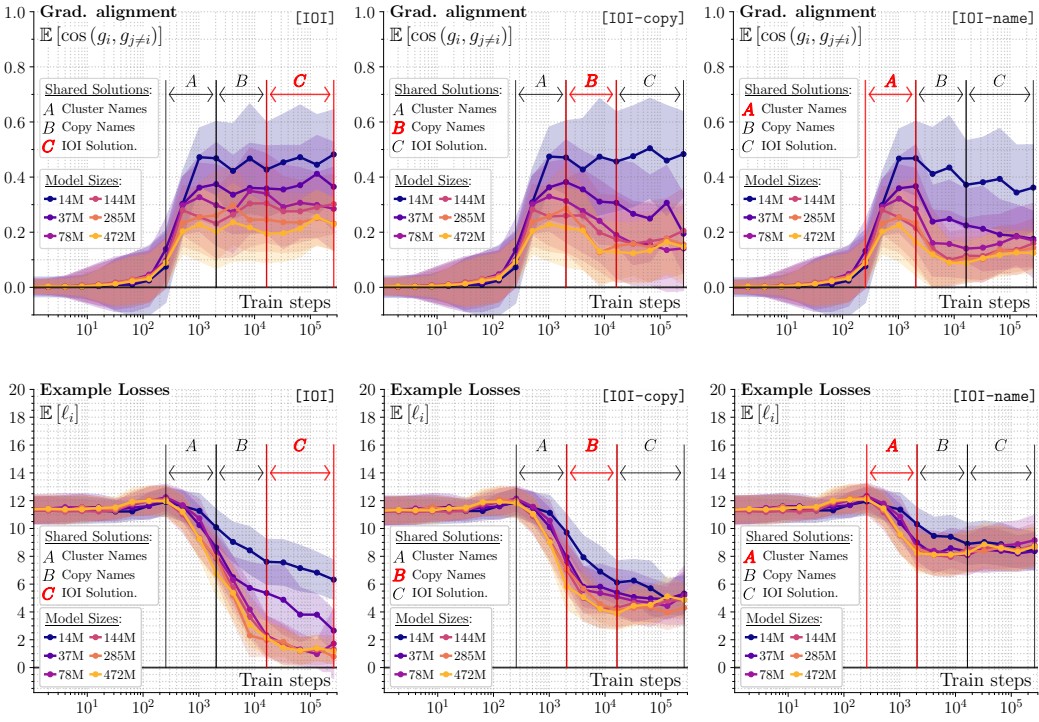

Figure 3: Models learn and build on simpler shared solutions in IOI circuit emergence.

**RESULT 3:** *Models learn and build on simpler shared solutions in IOI circuit emergence*

Plotting losses and gradient alignment for **[IOI]** along our **[IOI-name]** and **[IOI-copy]** controls in Figure 3 reveals a set of phenomena that further corroborate our hypothesis on one hand, and improve our mechanistic understanding of circuit emergence from Tigges et al. (2024) on the other.

**[IOI-copy]** has the shared solution of finding and copying previous names in a prompt, corresponding to the *Name Mover Head* that Wang et al. (2023) identify as a key subgraph in IOI circuits. We find that improvements in loss and gradient alignment overlap significantly with **[IOI]** during initial shared gradient discovery, suggesting that the initial shared solutions closely overlap for **[IOI]** and **[IOI-copy]**. However, near 20k steps, **[IOI-copy]** and **[IOI]** diverge, with losses no longer improving on **[IOI-copy]** and gradient alignment going down for all but the smallest models.

Conversely, **[IOI-name]** has the simpler but less effective shared solution of predicting every name in the **[IOI]** dataset with uniform probability. In Appendix B.3, we show that this solution corresponds to semantic clustering of name unembeddings in models' output heads. Similar to **[IOI-copy]**, we see initially that improvements in loss and gradient alignment overlap significantly with those of **[IOI]** and **[IOI-copy]** before similarly diverging, earlier this time, near 4k steps.

These results suggest that shared solutions for **[IOI-copy]** and **[IOI-name]** have *shared gradient features* that overlap with and generalize to **[IOI]**. Whether these persist throughout training or vanish after the observed divergences remains an open question. In Appendix B.4, we show how the emergence of IOI capabilities can be explained in terms of shared gradient features, and how gradient superposition can explain why smaller models struggle on **[IOI]** despite high gradient alignment.

## CONCLUSION AND FUTURE RESEARCH

Our findings are consistent with the hypothesis that shared gradient discovery & superposition underlies generalization in LLMs. However, further validating and characterizing this relationship remains an important direction for future research. In particular, developing a more comprehensive theoretical framework relating shared gradients to generalization, better accounting for and measuring gradient superposition in shared gradients, and extending our experiments beyond circuits to other instances of generalization are all important gaps not addressed in our work.

ACKNOWLEDGMENTS

We are grateful for support from IVADO and the Canada First Research Excellence Fund [EL]; the Canada CIFAR AI Chair Program and Canada Excellence Research Chairs Program [IR]; the NSERC post-graduate doctoral (PGS D) scholarship [AM]. IR and AM are also supported by the Simons Collaboration on Physics of Learning and Neural Computation (SFI-MPS-POL-00012574-09). We would also like to thank Mila (`mila.quebec`) and its IDT team for providing and supporting the computing resources used in this work.

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

# A  Background & Related Works

## A.1  Learning Dynamics of Generalization in LLMs

Generalization, the ability to make accurate predictions on previously unseen data (Bousquet & Elisseeff, 2002; Smith & Le, 2018), is a crucial property of machine learning algorithms. Together with emergent abilities (Wei et al., 2022), such as advanced reasoning and problem-solving, the strong generalization capabilities of large language models have made them highly valuable across a wide range of real-world applications. Existing research has examined the interplay between generalization and memorization (Wang et al., 2024; Kang et al., 2025), how LLMs acquire knowledge (Chiang et al., 2020; Wilson et al., 2023; Chang et al., 2024; Zhu et al., 2025b) and memorize training data (Carlini et al., 2021; Morris et al., 2025), and their performance across tasks of varying difficulty (Zhang et al., 2024; Kordi et al., 2025). Several studies have also investigated LLM generalization from a theoretical perspective (Lotfi et al., 2024; Finzi et al., 2024; Huang et al., 2024b).

In addition, a growing body of work focuses on the learning dynamics of LLMs. While some studies analyze dynamics during pre-training (Saphra & Lopez, 2019; Tirumala et al., 2022; Mircea et al., 2025; Li et al., 2025; Springer et al., 2025; Fehlauer et al., 2025), others concentrate on fine-tuning (Goyal et al., 2022; Ren & Sutherland, 2024; Lampinen et al., 2025; Li et al., 2025) and in-context learning (Nguyen & Reddy, 2024; Lampinen et al., 2025; Dherin et al., 2025). An alternative lens on training dynamics involves analyzing loss landscape sharpness (Kalra et al., 2026), which has informed the design of optimizers better suited for practical LLM training Huang et al. (2024a); Wang et al. (2025); Song et al. (2025). Despite significant progress in explaining the capabilities of LLMs, a more comprehensive understanding of the underlying mechanisms, integrating perspectives from feature learning, mechanistic interpretability, and optimization, remains an open and important research direction.

## A.2  Circuits and Circuit Emergence in LLMs

Circuits (Olah et al., 2020; Elhage et al., 2021) are computational subgraphs responsible for task-specific behavior in neural networks (Shi et al., 2024). Their primary advantage over other mechanistic interpretability approaches is that they identify the network components implementing concrete algorithms that drive model predictions. In this sense, circuits provide a complete explanation of how inputs are transformed into outputs. Consequently, studying how circuits emerge and evolve during training is essential for understanding the generalization capabilities of LLMs. Prior works have examined the individual components that constitute circuits (Olsson et al., 2022; Wang et al., 2023; Gould et al., 2023; McDougall et al., 2024; Singh et al., 2024; Ahmad et al., 2025), the extent to which circuits generalize across related tasks (Merullo et al., 2023; Nainani et al., 2024; He et al., 2025), their emergence over the course of training (Tigges et al., 2024; Hakimi et al., 2025), and the influence of specific data samples on circuit formation (Chen et al., 2026). The acquisition of new knowledge is also a central theme in continual learning, where understanding the mechanisms of circuit formation is crucial for adapting models to new data while mitigating catastrophic forgetting (Zheng et al., 2024a;b; Ou et al., 2025; Yao et al., 2026). Moreover, Prakash et al. (2023) show that fine-tuning LLMs on mathematical data strengthens the entity tracking circuit, consistent with our intuition on gradient destructive interference and shared gradients.

Despite this progress, the formation of circuits in LLMs from an optimization perspective remains poorly understood. The relationship between circuit learning and generalization has primarily been studied in smaller models. In particular, the phenomenon of grokking, characterized by a sudden transition to strong generalization after achieving zero training error (Power et al., 2022), has been explained through competition between memorizing and generalizing circuits (Nanda et al., 2022; Merrill et al., 2023; Varma et al., 2023; Huang et al., 2024c). Additionally, Chrisman et al. (2025) identify low-rank subnetworks, corresponding to specific directions in the parameter space, that can effectively reconstruct the gradients for individual training examples. Thus, studying how gradient updates drive circuit learning is crucial for understanding the emergence of generalization in LLMs.

## A.3 Generalization as Shared Gradient Discovery

The idea that shared gradient directions play a central role in generalization has been extensively studied in settings that require strong performance across multiple tasks, such as multi-task learning (Yu et al., 2020; Wang et al., 2020; Lee et al., 2021; Liu et al., 2021), continual learning (Lopez-Paz & Ranzato, 2017; Gupta et al., 2020), meta-learning (Nichol et al., 2018; Eshratifar et al., 2018; Guiroy et al., 2019; Chang & Lipson, 2023), and domain generalization (Shi et al., 2021; Phan et al., 2024). Several proposed methods explicitly regulate gradient alignment across tasks (Lopez-Paz & Ranzato, 2017; Yu et al., 2020; Wang et al., 2020), and some go further by directly maximizing this alignment (Nichol et al., 2018; Eshratifar et al., 2018; Shi et al., 2021; Phan et al., 2024). Moreover, Sankararaman et al. (2020) show that misaligned gradients slow down the convergence of overparameterized models.

However, the phenomenon of zero-sum learning (Mircea et al., 2025) suggests that creating shared gradient directions through optimization in LLMs is non-trivial, even when training is limited to a single objective such as next-token prediction. The river–valley perspective of the loss landscape (Wen et al., 2024) provides an interpretation in which the river direction loosely corresponds to shared gradient directions. Nevertheless, this viewpoint primarily emphasizes optimization dynamics, while the impact of shared gradients on generalization and feature learning in LLMs is not systematically investigated. In a relevant work following this direction, Parascandolo et al. (2020) experiment with small models and show that finding shared directions in per-example gradients leads to learning shared data-explaining mechanisms, which is key to generalizable and robust models.

## A.4 Superposition in Gradients

Superposition (or polysemanticity) (Mu & Andreas, 2020; Elhage et al., 2022) refers to the phenomenon in which individual neurons respond to multiple patterns in data, enabling neural networks to represent a number of features that far exceeds the dimensionality of their representations. Both theoretical (Hänni et al., 2024; Liu et al., 2025) and empirical (Hu et al., 2025; Bereska et al., 2025; Aravindan et al., 2025) studies have demonstrated the presence of superposition, showing that it arises even in simple toy models (Elhage et al., 2022; Scherlis et al., 2022). Beyond its representational implications, superposition has motivated a range of mechanistic interpretability approaches (Fong & Vedaldi, 2018; Bricken et al., 2023), which seek to disentangle the multiple features encoded within single neurons into more interpretable components.

From a generalization perspective, Morcos et al. (2018) demonstrate that superposition correlates with reduced memorization and better test performance. At the same time, superposition in neural activations emerges during training (Xu et al., 2025; Zhu et al., 2025a), suggesting that it is implicitly reflected in the gradients that drive weight updates. Despite this, most interpretability research has focused on explaining internal representations, leaving the role of superposition in gradients largely unexplored. Some distantly related works manipulate gradients to localize knowledge within specific subnetworks (Cloud et al., 2024; Shilov et al., 2025), effectively eliminating superposition in gradient signals. In this paper, we aim to draw attention to this gap in the literature by establishing a connection between shared-gradient discovery and superposition. We hope that this perspective helps bridge mechanistic interpretability and optimization research in large language models, and informs future work on understanding generalization and improving training dynamics.

# B  ADDITIONAL DETAILS & RESULTS FOR IOI CIRCUIT EMERGENCE

## B.1  IOI CIRCUIT DATASET

We use an Indirect Object Identification ([IOI]) dataset of 1000 examples, based on the original implementation[1]. The dataset is constructed from templates of the form *'Then, [B] and [A] went to the [PLACE]. [B] gave a [OBJECT] to '*, where *[A]* and *[B]* denote name tokens and *[PLACE]* denotes a place token, all drawn from predefined entity lists. The model's task is to predict the token *[A]* as the continuation of this prefix. In addition, we compare the observed loss values and gradient alignments to corresponding values for non-IOI token predictions, which we refer to as *controls*. Specifically, we consider the following control conditions:

- **[IOI-rand]**, random control: *[A]* is replaced with an arbitrary random token.
- **[IOI-name]**, cluster control: *[A]* is replaced with an arbitrary name token.
- **[IOI-copy]**, copy control: *[A]* is replaced with the second, non-IOI name token, i.e., *[B]*.

## B.2  GRADIENT ALIGNMENT IN RANDOM CONTROL

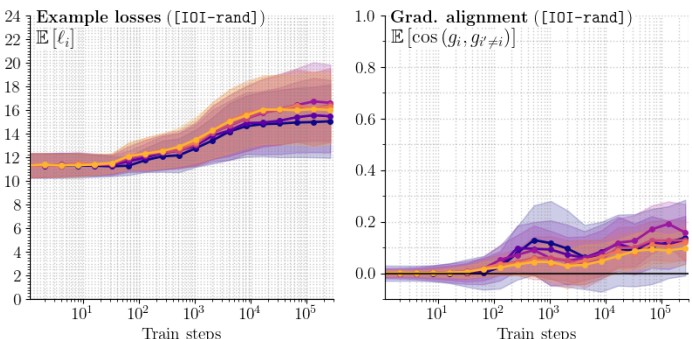

Figure 4: Gradient alignment and learning in IOI-rand control

## B.3  SEMANTIC CLUSTERING OF UNEMBEDDINGS AS EARLY SHARED SOLUTIONS

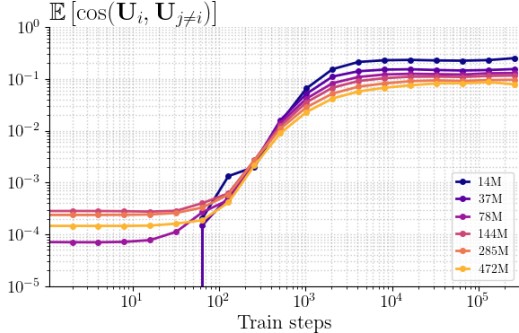

Figure 5: Output layer unembeddings for names in the **[IOI]** dataset, $U_i$, become clustered in a period that corresponds to learning and shared gradient discovery for **[IOI-name]** examples.

---

[1] https://github.com/redwoodresearch/Easy-Transformer/blob/main/easy_transformer/ioi_dataset.py

## B.4 WHY SMALLER MODELS FAIL TO LEARN IOI DESPITE GRADIENT ALIGNMENT

When measuring and plotting gradient alignment between examples from IOI and examples from IOI controls, we see that the overlap in shared gradient discovery from Figure 3 indeed corresponds to shared solutions between controls. In other words, shared gradient discovery is not independent between controls, and IOI examples have shared gradient features that overlap with examples from the controls.

Interestingly, we see alignment between IOI and controls go down at the same time as alignment within controls goes down. However, unlike alignment within controls, we see that alignment between IOI and controls becomes negative (albeit with high variance), but not for the smaller models that struggle to learn IOI. A likely explanation for this is that learning IOI at that point involves learning gradient features that are no longer shared with the copy or name controls, however smaller models do not have the capacity to learn these new gradient features in superposition with existing features.

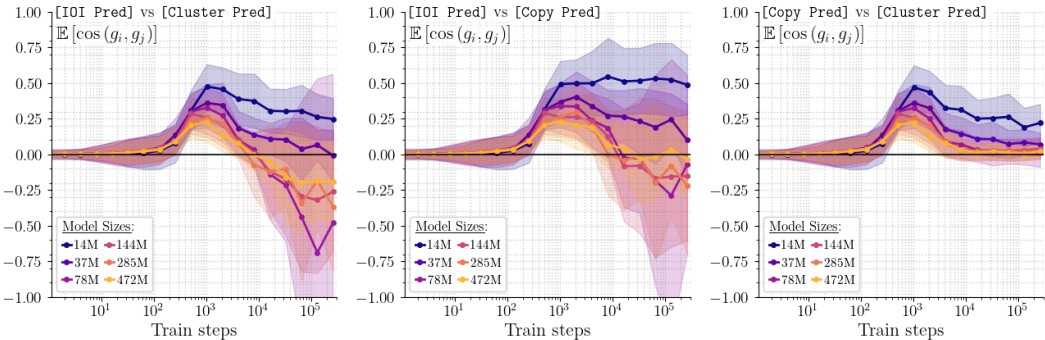

Figure 6: Gradient alignment between IOI and controls

## C    Experiments on Additional Circuits

In this Appendix section, we present results for two additional circuits: the Greater Than (GT) and One Month Later (OML) circuits. Overall, the observations from these circuits are consistent with and complement those reported for the IOI circuits in the main text, further supporting our findings.

### C.1    Circuit Datasets

**Greater Than (GT) Circuit**    The Greater Than ([GT]) circuit, proposed by Hanna et al. (2023), serves as a simple probe of a model's numerical reasoning abilities. The model is presented with prefixes such as *'The war lasted from the year 1732 to the year 17'* and is expected to predict a valid two-digit ending year greater than *32*. We use the dataset of 1000 examples provided by Hanna et al. (2024)[2]. Similarly to the IOI dataset, we compare our observations to a set of controls:

- **[GT-year]**: valid end year is replaced with an arbitrary two-digit token.
- **[GT-less]**: valid end year is replaced with a two-digit token corresponding to an earlier year (i.e., *<32* in the example above).
- **[GT-copy]**: valid end year is replaced with the year from the prefix (i.e., *32* in the example above).

**One Month Later (OML) Circuit**    We introduce a One Month Later ([OML]) circuit, inspired by the GT circuit but designed to be easier for smaller models to learn. This dataset tests whether a model has learned the sequential ordering of months within a year. Given a prefix such as *"The effect lasted one month, from the month of November to the month of"*, the model is expected to predict *December*. Since the original GT dataset contains only 34 unique events (i.e., templates), the OML dataset expands this structure by instantiating all 12 months, yielding $34 \times 12 = 408$ unique examples, compared to the random sample of 1000 unique examples used in the GT dataset. As for the controls, we use the following examples:

- **[OML-month]**: valid month is replaced with an arbitrary month token.
- **[OML-copy]**: valid month is replaced with a month token from the prefix (i.e., *November* in the example above).

---

[2]`https://github.com/hannamw/EAP-IG/blob/main/greater_than_data.csv`

## C.2 GREATER THAN (GT) CIRCUIT

Similar to our results for IOI, we observe an initial rise in shared gradients that co-occurs with a drop in loss across controls. However, unlike IOI, we observe gradient alignment dropping for all controls as well as the correct GT prediction. Furthermore, while gradient alignment does not appreciably increase after that for GT predictions, it does so for GT-copy and GT less predictions. At the same time, we see that loss and probabilities for correct GT predictions improve while those for incorrect GT-copy and GT-less predictions degrade despite increasing gradient alignment. These results suggest that the model is learning a shared solution in the form of suppressing the prediction of incorrect less-than and copy predictions rather than directly increasing the probability of correct greater-than prediction. These results are distinct from those observed in IOI but suggest a similar underlying phenomenon of learning shared solutions by discovering shared gradients. However, in this case, the shared solutions correspond to incorrect predictions and their suppression, hence the counter-intuitive results.

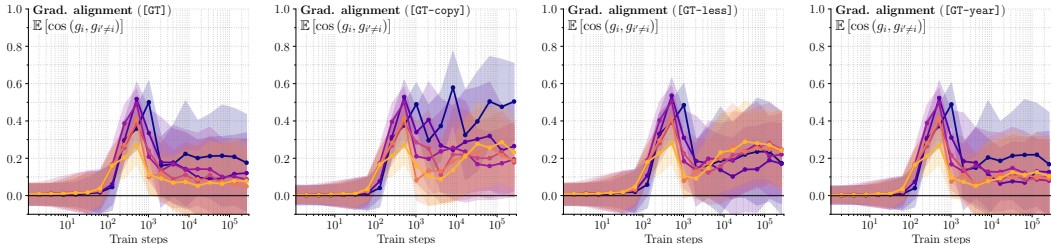

Figure 7: Example gradient alignment in GT and GT control datasets.

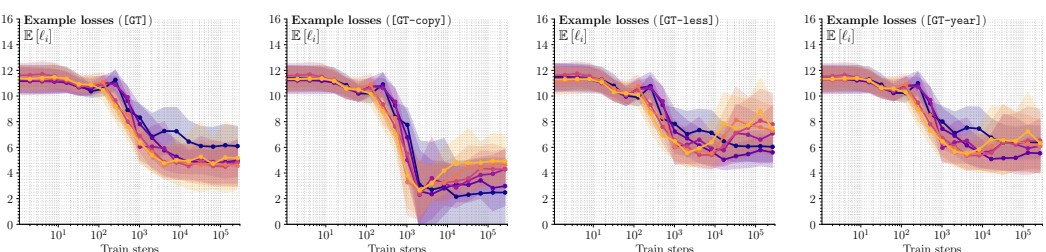

Figure 8: Example losses in GT and GT control datasets.

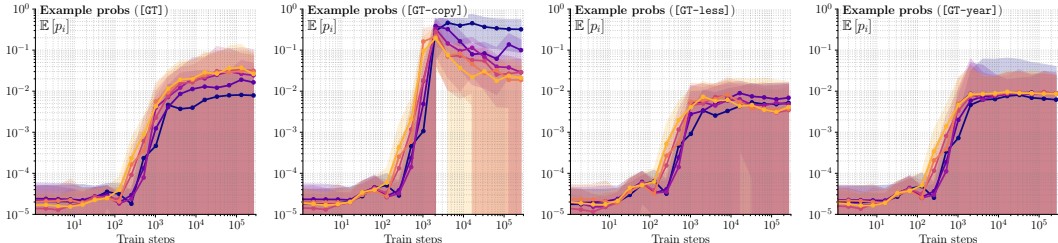

Figure 9: Example probability (of single year) in GT and GT control datasets.

## C.3 ONE MONTH LATER (OML) CIRCUIT

Similar to GT and IOI, we again observe an initial increase in gradient alignment that co-occurs with loss improvements and is shared across controls. Again, we observe that gradient alignment goes down for different controls when loss stops improving. Despite the simpler nature of this dataset, models still struggle to learn the correct prediction, similar to GT. Also similar to GT, we see evidence that a large part of learning OML boils down to suppressing incorrect solutions, specifically copying in this case. However, unlike GT, we observe that gradient alignment does go back up for correct OML predictions after the initial drop. At the same time, we see loss and probabilities of correct predictions improve beyond what can be explained from e.g. copy suppression and the decreasing probability of copy predictions. In other words, it seems that models are teetering towards learning a shared solution for OML predictions; although, unlike IOI, we do not conclusively observe this at the model and training scales of our experiments.

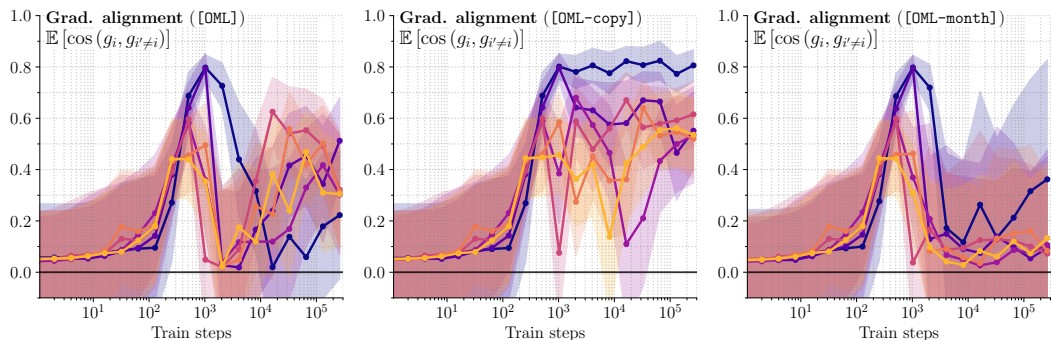

Figure 10: Example gradient alignment in OML and OML control datasets.

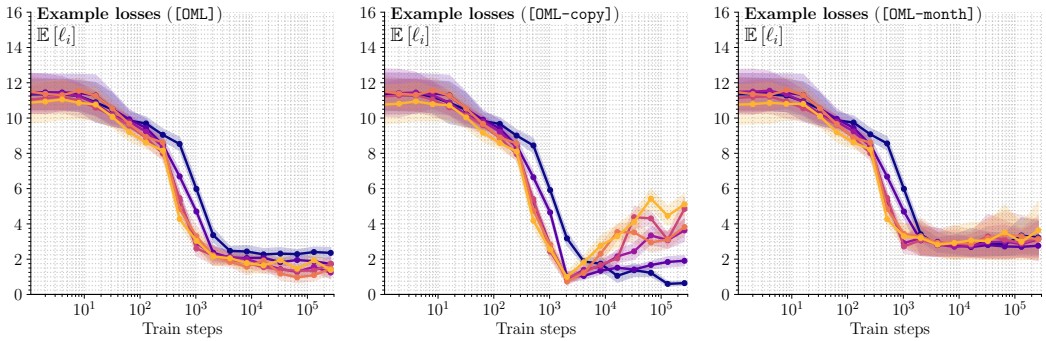

Figure 11: Example losses in OML and OML control datasets.

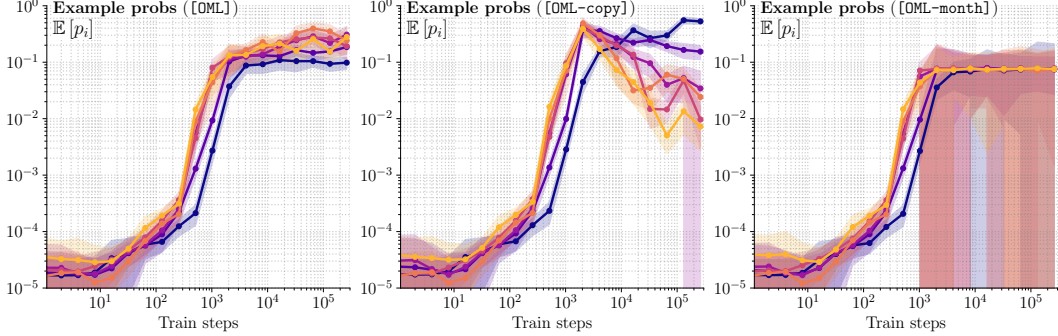

Figure 12: Example probability (of single month) in OML and OML control datasets.

# D  METHODOLOGICAL DETAILS

## D.1  MODELS

In our experiments, we use the language model and optimizer checkpoints published[3] by Mircea et al. (2025). These are based on the OLMo codebase (licensed under Apache-2.0) and trained with the publicly available training dataset of OLMo-7B-0724 Groeneveld et al. (2024). Model dimensions and learning rates are shown in Table 1. Training runs are $2^{18}$ ($\approx$262K) steps with a batch size of 0.5M tokens. Trapezoidal learning rate scheduling is used with a learning rate warmup to the values in Table 1 in the first 2000 steps and no cooldown in the $2^{18}$ steps considered.

Table 1: Model and Optimizer Parameters for Models

| Model size | 14M | 37M | 78M | 144M | 285M | 472M |
|---|---|---|---|---|---|---|
| d_model | 256 | 512 | 768 | 1024 | 1536 | 2048 |
| mlp_dim | 256 | 512 | 768 | 1024 | 1536 | 2048 |
| n_heads | 4 | 8 | 12 | 16 | 16 | 16 |
| n_layers | 4 | 8 | 12 | 16 | 16 | 16 |
| peak_lr | 1.3E-3 | 9.7E-4 | 8.0E-4 | 6.8E-4 | 5.7E-4 | 4.9E-4 |
| warmup | 2000 | 2000 | 2000 | 2000 | 2000 | 2000 |

## D.2  MEASURING SHARED GRADIENTS

To tractably measure pairwise cosine similarities between example gradients, we override the backward pass of a PyTorch linear layer (Code Snippet 1), using a buffer to allow for partially computing chunks of pairwise similarities when not all examples can fit in memory. This implementation assumes each example is one sequence and does not compute pairwise similarities between token gradients in the same sequence. Furthermore, our models use non-parametric layer normalization so that, except embedding weights, there are no other weights besides those in linear layers. However, computing pairwise gradient similarities for embedding and unembedding layers is computationally intractable because of the large vocabulary dimension in language models. Therefore we only compute and report gradient alignment for non-embedding parameters, which early experiments showed to be consistent with using the full parameters.

---

[3] https://github.com/mirandrom/zsl

```python
import torch
import torch.nn.functional as F

class LinearFunction(torch.autograd.Function):
    # Dimensions:
    # X / grad_X: (b,x) inputs
    # Y / grad_Y: (b,y) outputs
    # W / grad_W: (y,x) weights

    @staticmethod
    def forward(ctx, X, W):
        ctx.save_for_backward(X, W)
        Y = F.linear(X, W)
        return Y

    def backward(ctx, dldy):
        x, W  = ctx.saved_tensors

        # Compute GAA metrics and update buffers
        dldW = torch.einsum("BSx,BSy->Byx", x, dldy)
        update_gaa_buffers(
            grads=dldW.flatten(1, 2),
            buffer_dotp=W.buffer_dotp,
            buffer_l2sq=W.buffer_l2sq,
            grads_slice=getattr(W, "grads_slice", None),
            buffer_slice=getattr(W, "buffer_slice", None),
        )

        grad_X = torch.einsum("b...y,yx->b...x", grad_Y, W)

        return grad_X, None

@torch.no_grad()
def update_gaa_buffers(
    *, grads, buffer_dotp, buffer_l2sq, grads_slice=None, buffer_slice=None
):
    """
    :param grads: (n,d) tensor of n flattened gradients
    :param buffer_dotp: (n,n) tensor to store pairwise dot products
    :param grads_slice: optional tuple of slices for partial computation
    :param buffer_slice: optional tuple of slices for partial computation
    """
    # Slice grads if needed
    # (allows computing only a block of pairwise dot products)
    if grads_slice is None:
        A = grads
        B = grads
    else:
        ((a1, a2), (b1, b2)) = grads_slice
        A = grads[a1:a2]
        B = grads[b1:b2]

    # Slice output buffer if needed
    # (allows caching blocks of pairwise dot products)
    if buffer_slice is None:
        dotp = buffer_dotp
        l2sq = buffer_l2sq
    else:
        ((a1, a2), (b1, b2)) = buffer_slice
        dotp = buffer_dotp[a1:a2, b1:b2]
        l2sq = buffer_l2sq[a1:a2]
    dotp.copy_(A @ B.T)
    l2sq.copy_(torch.einsum("id,id->i", A, A))
    return
```

**Code Snippet 1:** PyTorch code for computing pairwise gradient similarites in linear layer.
Note that we compute and cache pairwise gradient dot products and square norms and compute cosine similarities from these.

# E  SCIENCE OF DL IMPROVEMENT CHALLENGE SUBMISSION

## E.1  WHAT MODEL ARE YOU TARGETING?

*Provide a summary of the problem the deep net model is designed to solve. Good summaries should outline the state of the literature, provide an overview that domain experts would consider reasonable, and cite relevant sources.*

In this work, we study large language models (LLMs), focusing specifically on the OLMo variant Groeneveld et al. (2024) with the checkpoints provided by Mircea et al. (2025). We consider models across a range of scales, from 14M to 472M parameters, as increasing model size (Kaplan et al., 2020) is a key factor in the emergence of new capabilities (Wei et al., 2022), when models begin to exhibit advanced reasoning and problem-solving skills beyond a certain scale. Our analysis examines how the generalization properties of LLMs arise during pre-training, the most resource-intensive phase of training, during which the model learns to predict the next token given a prefix, using a large corpus of text. Following pre-training, large-scale models are further adapted to downstream tasks (Zhang et al., 2025) or deployed as dialogue agents (Yi et al., 2025). Our approach thus helps to trace the emergence of generalization to specific stages of pre-training and changes in model scale.

## E.2  HOW DO YOUR RESULTS CONTRIBUTE—OR COULD POTENTIALLY CONTRIBUTE—TO UNDERSTANDING THESE MODELS?

*What aspects of the models become better understood thanks to your work?*

Existing studies of LLMs generally focus on one of two perspectives: mechanistic interpretability, which investigates how the algorithms responsible for specific model behaviors are implemented within the network, and optimization, which seeks to understand how gradient updates drive the evolution of the loss function or internal representations. In this work, we aim to bridge these two research directions by examining how shared gradients lead to the emergence of generalization through circuit formation. Specifically, we present experimental evidence that generalization to structured concepts in the data, such as Indirect Object Identification, requires the discovery of shared gradient directions that align across training examples and correspond to those concepts. Furthermore, we formalize the concept of gradient superposition and relate it to the learning of solutions that simultaneously represent multiple concepts occurring in the training data.

## E.3  HOW DO YOU EXPECT YOUR SUBMISSION TO INFLUENCE FUTURE WORK?

*Propose ways in which your insights, findings, or methodologies could shape subsequent research directions, model design choices, or scientific applications.*

Our work provides a framework for studying different generalization mechanisms in LLMs through the lens of gradient optimization. Since understanding a black-box solution is a prerequisite for improving it, we anticipate that our framework will serve as a useful tool for researchers investigating LLM behavior. Potential future outcomes include: (1) a unified understanding of how LLMs acquire generalization capabilities, (2) methods for more efficient and effective LLM training, and (3) strategies for mitigating undesired behaviors, such as memorizing training data without true comprehension or generating inappropriate or biased responses.

