# OpenReview forum: "Shared Gradient Discovery and Superposition: Learning Dynamics of Generalization in LLMs"
_ICLR.cc/2026/Workshop/Sci4DL — Sci4DL 2026_

### Official Review · Reviewer_7rDJ · 2026-02-18

**Fit:** 3
**Significance:** 3
**Confidence:** 2

**Summary:**

This paper proposes a mechanism of study generalization in LLMs via the aspects of shared gradients. Specifically, for a sample $x_i$, the assumption the paper made is the loss $\ell_i$ is locally linearized around the parameters $\theta$ such that gradient descent on $x_i$ will also improve upon another sample $x_{i'}$ if the dot product between their gradients $\langle g_i, g_{i'} \rangle > 0$. Their results focus on the task of ***Indirect Object Identification*** which illustrate there is a plausible trend, involving aspects of shared gradients, showing the behavior of the model as it trains.

**Strengths:**

- Although the paper is very empirical, I think it is an interesting concept and the author(s) have provided a lot of results to back up their claims.

**Suggestions:**

- For your plots, perhaps you should switch the grid to plt.grid(..., which='major'). There are a lot of bars and they might be too distracting for others. Nonetheless, the plots are visually fine.

- So, your experiments are related to training steps and various model sizes. Any considerations of the training data sizes or how correlated the data points are?

- For theoretical aspects, perhaps take a look at the concept of [Pseudo-likelihood](https://en.wikipedia.org/wiki/Pseudolikelihood). I suspect that the reliance on modeling conditional likelihood for LLMs, likely reflect what pseudo-likelihood is doing. Specifically, for each token $x_i$, the Transformer learns this neighborhood around it where $p_\theta (x_i | x_{i'})$ represents the next-token prediction given a context (or neighborhood) around $x_i$.

---

### Official Review · Reviewer_dxSD · 2026-02-25

**Fit:** 3
**Significance:** 2
**Confidence:** 2

**Summary:**

The paper studies how generalization emerges in LLMs through the lens of shared gradients and superposition. The core idea is that, for an LLM to generalize correctly, different training examples must point in the same direction in the weight space: aligned gradients on different examples automatically improves the other. By analyzing several metrics (IOI, gradient alignments and losses), the authors show that, as models learn to complete sentences, they first discover simple shared solutions before developing more complex reasoning circuits.

**Strengths:**

This paper moves beyond the current 'bigger is better' trend to understand the underlying mechanics of how LLMs actually generalise. It is well-structured and provides compelling empirical evidences showing the synchronized emergence of shared gradient discovery and circuit formation. It also thoroughly cites previous and related work.

**Suggestions:**

**Comparative analysis of generalization vs failures**

While the paper successfully maps shared gradients discovery to the emergence of circuits, is it possible to similarly study identified generalization failures? For instance taking an out-of-distribution task or identify a case where the model memorized the answer. In such cases, do you think the failure occur because: 1) the lack of alignment; or 2) shared gradient exist but are 'killed' by interferences from other tasks?

**Characterization of a 'discovery' phase transition**

You identify that discovery does happen, but without focusing on when. Can your framework be used to study (at least empirically and perhaps on toy models) if shared gradient discovery follows a phase transition (similar to grokking for instance). This could be for instance identify a critical number of data or training time required for shared gradients to crystallize.

---

### Official Review · Reviewer_KkmR · 2026-02-27

**Fit:** 3
**Significance:** 2
**Confidence:** 2

**Summary:**

The paper claims that in order to achieve mechanistic understanding of the generalization capabilities of large language models (LLMs) one has to pinpoint the right model abstraction that will reduce the high-dimensional problem/system into low-dimensional interpretable components. To that end, the paper builds on the idea of  Mircea et al. (2025) that generalization in an LLM can be mechanistically linked to whether the gradients for each example across a set of examples have positive alignment and, more generally, to whether they share feature-aligned components under a factorization that assumes gradient superposition.
More precisely, the authors assume a local linearization of the loss function (cross entropy loss) and consider that examples in a set X have shared (aligned) gradients if the gradients of each pair of examples in the set have positive inner product, for all possible example pairs.
They name shared gradient directions the set of directions that have positive inner product with all per-example gradients in the set X.
They assume that example gradients can be factorized as linear combinations of near-orthogonal gradient features.
The mechanism for generalization is that under the assumption of a locally linearized loss around the model parameters theta, taking one gradient step for example $i$ will automatically end up on loss improvement also for example $j$ if the gradients for $i$ and $j$ are positively aligned.
The authors test this idea empirically  by tracking gradient-alignment metrics in the context of emergence of a known circuit (Indirect Object Identification) during network pretraining.

**Strengths:**

- The authors propose an interesting and elegant idea for mechanistic understanding of generalization in LLMs.
- The paper is well written with visualizations that help understand the introduced concepts.
- I like the idea of tracking gradient cosine similarity during circuit emergence, because it provides a concrete way to link training dynamics to circuits used for mechanistic interpretability.

**Suggestions:**

- The theoretical part of the paper posits that for a set with shared gradients, all gradients in the set should have aligned directions/positive inner products. This would imply that the existence of one pair with negative alignment would be enough to violate the definition. However in their numerical experiments they probe for high average cosine similarity (they report mean and std cosine similarity), which is not exactly the same as saying that there is no pair with negative alignment. For that they would have to demonstrate precisely .
- The author’s mention that they do not consider embeddings/unembedding layers and use only the non-embedding parameters. This essentially implies that they computed alignment of gradients only wrt to a subset of parameters, and not cot considering parameters that may directly influence output logins, like the unembedding layer. While the authors argue that early experiments were consistent, for a future publication I would expect this “consistency” to be explicitly quantified.
- In the numerical experiments the authors show that the alignment of gradients co-occurs with circuit emergence. However, they do not establish causality for this, they do not test whether gradient alignment is necessary or sufficient for the emergence of the IOI circuit.



- The proposed mechanism relies on the local linearizability of the loss function, but does not try to quantify when this approximation is valid or breaks. While it is an elegant theoretical idea, I personally do not have the experience to know whether this is a valid claim in practice and when or how the learning rate or the underlying curvature of the loss landscape interact to render this approximation poor.

- Do you think that if you would induce manually gradient alignment the IOI circuit might have emerged earlier? Have you tried something along these lines?
- For validating the central claim of the paper I would expect a more clear example that shows that gradient alignment leads to loss improvement.

#### Minor

- In the definition of Shared gradient I think there is an issue with the wording because in the current version of the beginning of the definition i appears to be the set of examples, while i is an example \in the set of examples X.

---

### Meta-Review · Area_Chair_hi4w · 2026-03-01

**Recommendation:** Accept

**Metareview:**

All reviewers posted positive reviews about this paper. I recommend accept.

---

### Decision · Program_Chairs · 2026-03-02

Accept